# The Anti-Tumor Effect of *Lactococcus lactis* Bacteria-Secreting Human Soluble TRAIL Can Be Enhanced by Metformin Both In Vitro and In Vivo in a Mouse Model of Human Colorectal Cancer

**DOI:** 10.3390/cancers13123004

**Published:** 2021-06-15

**Authors:** Katarzyna Kaczmarek, Jerzy Więckiewicz, Kazimierz Węglarczyk, Maciej Siedlar, Jarek Baran

**Affiliations:** Department of Clinical Immunology, Institute of Pediatrics, Jagiellonian University Medical College, Wielicka Str. 265, 30-663 Kraków, Poland or katarzyna.ciacma@gmail.com (K.K.); miwiecki@cyf-kr.edu.pl (J.W.); kazimierz.weglarczyk@uj.edu.pl (K.W.); misiedla@cyf-kr.edu.pl (M.S.)

**Keywords:** colorectal cancer, *Lactococcus lactis*, TRAIL

## Abstract

**Simple Summary:**

Colorectal cancer (CRC) is a major cause of morbidity and mortality in Europe, and accounts for over 10% of all cancer-related deaths worldwide. These indicate an urgent need for novel therapeutic options in CRC. Here, we analysed if genetically modified non-pathogenic *Lactococcus lactis* bacteria can be used for local delivery of human recombinant *Tumor Necrosis Factor-Related Apoptosis-Inducing Ligand* (TRAIL) and induction of tumor cells death in vitro and in vivo in CRC mouse model. We showed that modified *L. lactis* bacteria were able to secrete biologically active human soluble TRAIL (*L. lactis*(hsTRAIL+)), which selectively eliminated human CRC cells in vitro, and was further strengthened by metformin (MetF). Our results from in vitro studies were confirmed in vivo using subcutaneous NOD-SCID mouse model of human CRC. The data showed a significant reduction of the tumor growth by intratumor injection of *L. lactis*(hsTRAIL+) bacteria producing hsTRAIL. This effect could be further enhanced by oral administration of MetF.

**Abstract:**

*Tumor Necrosis Factor-Related Apoptosis-Inducing Ligand* (TRAIL) induces apoptosis of many cancer cells, including CRC cells, being non-harmful for normal ones. However, recombinant form of human TRAIL failed in clinical trial when administered intravenously. To assess the importance of TRAIL in CRC patients, new form of TRAIL delivery would be required. Here we used genetically modified, non-pathogenic *Lactococcus lactis* bacteria as a vehicle for local delivery of human soluble TRAIL (hsTRAIL) in CRC. Operating under the Nisin Controlled Gene Expression System (NICE), the modified bacteria (*L. lactis*(hsTRAIL+)) were able to induce cell death of HCT116 and SW480 human cancer cells and reduce the growth of HCT116-tumor spheres in vitro. This effect was cancer cell specific as the cells of normal colon epithelium (FHC cells) were not affected by hsTRAIL-producing bacteria. Metformin (MetF), 5-fluorouracil (5-FU) and irinotecan (CPT-11) enhanced the anti-tumor actions of hsTRAIL in vitro. In the NOD-SCID mouse model, treatment of subcutaneous HCT116-tumors with *L. lactis*(hsTRAIL+) bacteria given intratumorally, significantly reduced the tumor growth. This anti-tumor activity of hsTRAIL in vivo was further enhanced by oral administration of MetF. These findings indicate that *L. lactis* bacteria could be suitable for local delivery of biologically active human proteins. At the same time, we documented that anti-tumor activity of hsTRAIL in experimental therapy of CRC can be further enhanced by MetF given orally, opening a venue for alternative CRC-treatment strategies.

## 1. Introduction

Colorectal cancer (CRC) represents the group of gastrointestinal cancers (GI) and includes malignant tumors of the colon and rectum. It has a significant contribution in the global cancer incidence (10.2% CRC from 26.3% for all GI) and cancer-related mortality (9.2% CRC from 35.4% for all GI) in 2018 [1]. In Europe CRC causes a high social burden, with the second most frequent cancer and cause of cancer-related death [1,2]. The risk factors for CRC include genetic predispositions [3] and aberrant epigenetic alternations [4] from one side, and environmental factors, including alcohol consumption [5,6], high-fat diet [7,8], lack of physical activity [9] and chronic inflammation of intestinal mucosa [10], from the other. Currently, alterations in the composition of gut microbiome and its metabolites are also considered as risk factor for CRC development [11] and recently gained clinical interest as potential biomarkers for CRC screening and prognosis [12,13]. In this context, the most important are anaerobic *Fusobacterium* sp. and *Porphyromonas* sp. with proven pro-tumorigenic activity [14,15], while supplementation of diet with probiotics of *Lactobacillus* or *Bifidobacterium* sp., has been shown to reduce the risk of CRC [16,17,18].

One of the major problems with CRC is that it is usually diagnosed in the advanced stage, partly due to a lack of the public awareness for the need for preventive colonoscopy. Surgical removal of the tumor remains the main form of CRC treatment, followed by adjuvant chemotherapy, while for metastatic CRC chemotherapy remains the primary treatment option [19]. However, cytostatics are non-selective in their action towards rapidly dividing cells, causing severe side effects. In addition, available chemotherapy is still ineffective to cure patients with advanced CRC [20,21,22], while patients often acquire resistance for the treatment [23]. For patients with strictly selected genetic profiles, immunotherapy or molecular-targeted personalized therapy might also be included [24,25]. These options, however, are available only for carefully selected subgroups of patients. In this context the research aiming at developing new widely effective therapeutic approaches are crucial. Tumor implantation mouse models, with subcutaneous graft of human cancer cells, are very common for screening the candidate therapeutics [26].

*Tumor Necrosis Factor-Related Apoptosis-Inducing Ligand* (TRAIL) has been considered as a promising agent for anti-tumor therapy for a long time, as cancer cells show higher sensitivity to TRAIL comparing to normal cells, highlighting TRAIL’s potential as a novel and effective anti-cancer drug. However, many tumors turned out to be resistant for TRAIL-based treatment. Therefore, new TRAIL formulations and combined therapy models are seeking to enhance its bioactivity and effectiveness in vivo. One of such approach is using the genetically modified bacteria as vector and producer of human TRAIL [27,28,29] and agents potentiating its antitumor activity [30,31,32,33,34,35,36,37]. *Lactococcus lactis* is a non-pathogenic, Gram-positive, lactic acid bacterium (LAB), for years used in the dairy and pharmaceutical industry. Recently, it gained more interest also in the bio-medical research. *L. lactis* bacteria were the first microorganisms used in clinical studies in humans, where patients with moderate and advanced Crohn’s disease after treatment with genetically modified *L. lactis* producing human IL-10 showed in majority an improvement in clinical parameters assessed by Crohn’s Disease Activity Index (CDAI) [38]. Since LAB are commonly used as probiotics and some of the species belong to the natural microflora of the gut’s ecosystem, *L. lactis* might be a promising vehicle for local and safe delivery of therapeutic proteins in intestinal diseases, including CRC. A big step forward in this field was the development of a strictly-controlled and easy-to-operate system for the expression of heterologous genes in *L. lactis*—Nisin Controlled Gene Expression System (NICE^®^) [39,40]. Based on this advancement, we recently developed *L. lactis* NZ9000 bacteria, genetically modified to secrete human soluble TRAIL (hsTRAIL) and documented biological activity of this protein in the selective elimination of HCT116 human colon cancer cells in vitro via apoptosis [41].

Here we asked, if *L. lactis*(hsTRAIL+) bacteria would be able to release an active hsTRAIL locally in the tumor and, if so, what will be the efficacy of such a treatment. To address these questions, we used at first the in vitro co-culture system, followed by subcutaneous NOD-SCID mouse model of human CRC xenotransplant. Our results showed, that *L. lactis*(hsTRAIL+) bacteria secreted hsTRAIL in a co-culture with human CRC cells in vitro and intratumorally in vivo in subcutaneous HCT116-tumors, leading to the death of cancer cells and reduction of the tumor growth. The therapeutic effect of hsTRAIL could be further enhanced by metformin (MetF). These results document potential use of *L. lactis* bacteria as a vehicle for CRC targeting proteins and, at the same time, indicate MetF as an agent able to enhance tumoricidal effect of hsTRAIL in CRC treatment.

## 2. Materials and Methods

### 2.1. Study Design

Particular stages of the study are briefly presented in Appendix A. Development of genetically modified derivative of *L. lactis* NZ9000 strain, capable of an efficient hsTRAIL secretion (designated as “*L. lactis*(hsTRAIL+)”), and selective anti-tumor activity of such hsTRAIL through apoptosis, has been previously shown [41]. In the present research, biological activity of *L. lactis*-derived hsTRAIL was examined in more details, including in vitro co-culture of the bacteria with human CRC cells in a monolayer and 3D (spheres) system, and in vivo, using subcutaneous xenotransplant model of human CRC.

### 2.2. Cell Cultures

Human colon carcinoma cell lines HCT116 (CCL-247^TM^), SW480 (CCL-228^TM^) and human colon epithelium cell line FHC (CRL-1831^TM^, used as control) were obtained from the American Type Culture Collection (ATCC, Manassas, VA, USA). All the cell lines were maintained according to the distributor’s instructions, at 37 °C in humified atmosphere with 5% CO_2_. Briefly, HCT116 cells were cultured in McCoy’s 5A (Gibco, Paisley, UK) supplemented with 10% fetal bovine serum (FBS; Gibco) and gentamicin (50 µg/mL; Gibco). SW480 cells were cultured in Dulbecco’s Modified Eagle’s (DMEM; Sigma Aldrich, Saint Louis, MI, USA), supplemented with 10% FBS and 50 µg/mL of gentamicin. FHC cells were grown in DMEM:F12 (Gibco) supplemented with 10% FBS, 10 mM HEPES (Sigma Aldrich), 10 ng/mL cholera toxin (Sigma Aldrich), 0.005 mg/mL insulin (Sigma Aldrich), 0.005 mg/mL transferrin (Sigma Aldrich), 100 ng/mL hydrocortisone (Sigma Aldrich, USA) and gentamicin (50 µg/mL; Gibco). All the cells were passaged twice a week with 0.05% trypsin (PAN-Biotech GmbH, Aidenbach, Germany) in EDTA (Eurx, Gdansk, Poland) and regularly tested for *Mycoplasma* sp. contamination by PCR-ELISA test (Roche, Mannheim, Germany), according to manufacturer’s instruction.

### 2.3. Bacteria

*Lactococccus lactis* NZ9000 host strain was obtained from MoBiTec (Goettingen, Germany). Genetically modified *L. lactis* clones harbouring secretion plasmid vector pNZ8124 (MoBiTec; designated as “*L. lactis*(“empty” vector)”) and its modified derivative with human soluble TRAIL-cDNA placed downstream of the inducible promoter PnisA (*L. lactis*(hsTRAIL+)), were prepared as previously described [41] and cultured at 30 °C, without aeration, in M17 medium (BTL, Łodz, Poland) supplemented with 0.5% glucose and 10 µg/mL chloramphenicol (Cm10; Sigma Aldrich) to maintain the plasmid. Bacteria stocks were frozen in M17 medium, supplemented with 0.5% glucose, Cm10 and sterile 20% glycerol (Eurx) and stored at −80 °C until further use.

### 2.4. Induction of hsTRAIL Expression

According to the previously established culture conditions for most efficient hsTRAIL-expression [41], the M17 broth supplemented with 0.5% glucose and Cm10 was inoculated with *L. lactis*(hsTRAIL+) or negative control (*L. lactis*(“empty” vector)) at dilution 1:50 and grown overnight (ON) at 30 °C, without aeration. The ON cultures were subsequently diluted 1:50 in M17 broth medium supplemented with 0.5% glucose and Cm10 and grown once more ON at the same conditions. At the day of induction, the ON cultures were diluted 1:20 in M17 broth medium supplemented with 0.3% glucose, 0.3% L-arginine, ZnSO_4_ (100 µM), Cm10 and grown for additional 3 h at 30 °C without aeration, until the optical density (OD_600_) of the cultures reached 0.3–0.4. Then, the cultures were centrifuged for 30 min at 2800× *g* at room temperature (RT) and obtained cell pellets were precisely resuspended in a ¼ volume of “complete medium” (“CM”; M17 broth medium supplemented with 0.3% glucose, 0.3% L-arginine, 100 µM ZnSO_4_, serine proteases inhibitor- aprotinin (2 µg/mL; Bioshop, Burlington, Canada) and nisin (25 ng/mL; MoBiTec), as the inducer of hsTRAIL expression). The induction step was performed for total of 4 h at 30 °C, without aeration. Appendix A specifies the steps of *L. lactis*(hsTRAIL+) culture and optimized growth conditions.

### 2.5. Preparation of hsTRAIL+ Supernatants for In Vitro Studies

After 4 h incubation with nisin, the bacteria were centrifuged for 30 min at 2800× *g*, 4 °C and cell-free supernatant was collected. Next, the supernatant was concentrated (30 min at 2800× *g*), using the Thermo Scientific™ Pierce™ PES 10 K Protein Concentrator (Pierce Biotechnology, Rockford, IL, USA) and subsequently filtered through low protein-binding PVDF filters (Merck Millipore, Burlington, MA, USA), aliquoted and stored at −80 °C until further use. The concentration of hsTRAIL in the supernatants was measured by ELISA assay (LSBio^TM^, Seattle, WA, USA), according to the manufacturer’s instructions. The absorbance was measured at 450 nm and 570 nm (wavelength correction) using microplate reader ELx 800NB (Bio-Tec Instruments, Winooski, VT, USA).

### 2.6. MTS Assay

Viability of human cells after hsTRAIL treatment in vitro was assessed using the colorimetric method with tetrazolium compound 3-(4,5-dimethylthiazol-2-yl)-5-(3-carboxymethoxyphenyl)-2-(4-sulfophenyl)-2*H* (MTS) (CellTiter 96^®^ AQueous One Solution Cell Proliferation Assay, Promega, Madison, WI, USA), according to the manufacturer’s instruction. Briefly, after the treatment of cells, 20 µL of MTS reagent per 100 µL of the sample volume was added to the culture well and incubated in the dark for next 2 h in a 37 °C humified atmosphere with 5% CO_2_, until conversion of the dye to formazan, proportional to the number of living cells, occurred. The amount of formazan was estimated spectrophotometrically, by the absorbance measurement at 490 nm, using plate reader Spark^®^Tecan (Tecan, Mannedorf, Switzerland). The number of viable cells in each well was normalized to the control (cells incubated with standard cell culture medium, designated as “100% of viability”) and presented as % of control.

### 2.7. Co-Culture of L. lactis Bacteria with Human Cell Lines

The protocol for the co-culture of *L. lactis* bacteria with human colon cancer cells or normal colon epithelium cells was developed in own research. Briefly, HCT116, SW480 and FHC cells were harvested with trypsin and seeded onto flat-bottom 96-well plates (Sarstedt, Numbrecht, Germany) at a density 10^4^ cells/well in appropriate culture media supplemented, as described above, w/o antibiotics. After 20 h of culture in a 37 °C humified atmosphere with 5% CO_2_, the culture media were replaced with diluted bacteria, added at the induction phase.

*L. lactis*(hsTRAIL+) and negative control (*L. lactis*(“empty” vector) bacteria, were grown as described above. After 3 h of culture in M17 broth medium supplemented with 0.3% glucose, 0.3% L-arginine, ZnSO_4_ (100 µM), Cm10, at 30 °C without aeration, the bacteria were centrifuged (2800× *g*, 30 min, 21 °C) and washed once with a warm sterile phosphate-buffered saline (PBS; Corning, New York, NY, USA) supplemented with ZnSO_4_ (100 µM). The bacterial pellets were resuspended in appropriate cell culture medium enriched with 2% FBS, 0.1% L-arginine, 2 µg/mL aprotinin, 100 µM ZnSO_4_, to final dilutions 1/15; 1/20; 1/30, then induced with 25 ng/mL of nisin for hsTRAIL production and added directly to the plated cells. After 2.5 h of co-culture at 30 °C and humified atmosphere with 5% CO_2_, ampicillin (1 µg/µL; Sigma Aldrich) was added to stop the bacterial growth. The co-cultures were subsequently moved to 37 °C in humified atmosphere, 5% CO_2_ and incubated up to 48 h. The viability of human CRC cells or normal colon epithelial cell line was determined by the MTS assay (Promega). For this purpose, after 48 h of the co-culture, the cell-media containing *L. lactis*(hsTRAIL+) or *L. lactis* (“empty” vector) bacteria were removed and the cells were gently washed with a warm and sterile PBS to rinse out the rest of bacteria. Finally, the fresh culture media and MTS reagent were added. The measurement and final calculation of the % of viable HCT116, SW480 (Appendix A) and FHC cells after co-culture with the bacteria was performed, as described above.

### 2.8. Human Colon Cancer Cells Spheres

Spheres of human CRC cell line HCT116 were grown in a suspension, according to the manufacturer’s instructions (Cell2Sphere^TM^ kit; StemTek Therapeutics, Bilbao, Spain) on the provided plate, in the dedicated Cell2Sphere™ medium. Concentrated supernatants of the *L. lactis*(hsTRAIL+) broth culture were diluted in Cell2Sphere™ medium and added in the volume corresponding to a final hsTRAIL concentration of 50; 100; 1000 ng/mL. The equal volumes of the negative control (*L. lactis*(“empty” vector)) broth supernatants were run in parallel. As the control of viability and morphology, HCT116-spheres incubated in the Cell2Sphere™ medium alone were used. To assess the effect of hsTRAIL on the sphere formation, the supernatants of the culture medium were added to the spheres at the same day the HCT116-sphere culture was started, for following six days. To evaluate the effect of hsTRAIL-secreting *L. lactis* bacteria on HCT116-spheres, the bacteria were added on day 6th of the culture growing in a self-made “colorectal cancer-spheres medium” (CCSM; composition: DMEM:F12 (Gibco) supplemented with ITS Liquid Media Supplement, 5 mM HEPES, 4 mg/mL BSA, 3 mg/mL glucose (all from Sigma Aldrich), 2 nM L-glutamine (Gibco), 20 ng/mL eGF, 20 ng/mL bFGF (both from Thermo Fisher Scientific, Waltham, MA, USA), w/o any antibiotics) for the following 48 h.

*L. lactis*(hsTRAIL+) and negative control (*L. lactis*(“empty” vector)) bacteria were prepared for the co-culture with the HCT116-spheres, as described above (the co-culture with HCT116 cells growing in monolayer). After washing with sterile warm PBS supplemented with ZnSO_4_ (100 µM), the bacterial pellets were resuspended the CCSM medium enriched with 0.1% L-arginine, 2 µg/mL aprotinin, 100 µM ZnSO_4_, to final dilution 1/20 and subsequently induced with 25 ng/mL of nisin for hsTRAIL-expression, then added directly to the spheres. After 2.5 h of the co-cultures, ampicillin (1 µg/µL) was added to stop the bacterial growth and the co-cultures were moved to 37 °C in humified atmosphere, 5% CO_2_ and incubated up to 48 h.

After treatment with the supernatants from broth cultures or whole bacteria, the spheres were photographed using an OLYMPUS IX70 microscope (Olympus, Tokyo, Japan) and the compatible CellSens Dimension software (Olympus) was used to assess their morphology and further calculate their diameters. Measurements were done for four “optically the largest” spheres from each experimental group and their diameters were calculated using paint.net (version 4.1.6, Rick Brewster, Pullman, WA, USA). Then, MTS reagent (Promega) was added, and spheres viability was measured. The spheres viability after incubation in the presence of supernatants from *L. lactis* broth cultures was calculated by normalization to viability of the spheres in control (“Control(medium))”, viability 100%).

### 2.9. Selection of the Anti-Tumor Drug for the Combined Therapy with L. lactis(hsTRAIL+) Bacteria

The therapeutics and their respective doses for assessing the potential joined anti-tumor effect with *L. lactis*-derived hsTRAIL were selected according to the literature [30,31,32,33,34]. A stock of 1M metformin (MetF; Sigma Aldrich) was prepared in a sterile, deionized water (Eurx) and subsequently sterilized by filtration using a 0.22 µm PVDF membrane filter (Carl Roth, Karlsruhe, Germany). Stocks of 5-fluorouracil (5-FU; TriMen Chemicals, Lodz, Poland) and irinotecan (CPT-11; MedChem Express, Monmouth Junction, NJ, USA), at concentrations of 50 mg/mL, were prepared in DMSO (Sigma Aldrich). All therapeutic stocks were aliquoted, frozen, and stored at −20 °C until use. Before incubation with HCT116 cells, they were thawed and the following working dilutions were prepared (given at final concentrations): MetF—5; 10; 15; 20; 25 mM, 5-FU—0.192; 0.384; 0.768; 3.84; 7.68 µM, CPT-11—0.425; 0.85; 1.7; 2.125; 2.55 µM. A final concentration of 50 ng/mL of hsTRAIL, in the supernatant from *L. lactis*(hsTRAIL+) broth culture was chosen to investigate the potential joined anti-tumor activity of hsTRAIL and the therapeutics, since this concentration in our previous studies was shown to reduce viability of HCT116 cells to approximately 75% of control [41]. Additionally, HCT116 cells were subjected to MetF at the concentration of 20 mM with the supernatant from *L. lactis*(hsTRAIL+) broth culture, at the dilutions corresponding to increasing concentrations of hsTRAIL—10; 25; 50; 75; 100 ng/mL, or equal volumes of the supernatant from broth culture of bacteria harbouring ctrl vector (negative control). All samples were prepared in McCoy’s 5A medium supplemented with 2% FBS; for control cells only medium with FBS was added.

HCT116 cells were harvested with trypsin and plated on a 96-well plate at 10^4^ cells/well in McCoy’s 5A supplemented with 2% FBS. After 20 h of culture at 37 °C, 5% CO_2_ and 95% of humidity, the therapeutics alone, or in combination with the supernatant from broth culture of *L. lactis*(hsTRAIL+) bacteria, were added. Cells were maintained for the next 48 h at the same culture conditions. After the treatment, the supernatants were removed, replaced with fresh medium and MTS assay assessing cell viability was performed, as described above.

### 2.10. Animals for Subcutaneous CRC Model

To develop mice model of human CRC, the NOD-SCID (NOD.CB17/Prkdscid/ NCrHsd) mice, homozygous for the severe combined immune deficiency spontaneous mutation *Prkdc^scid^*, with the lack of mature B and T lymphocytes, as described previously [42,43] were chosen. NOD-SCID mice were bred and maintained in the local animal facility at the Department of Clinical Immunology, Jagiellonian University Medical College in Kraków, Poland. Mice were maintained in dedicated housing rooms under aseptic and controlled laboratory conditions: 12/12 h light/dark cycle, room temperature 20–22 °C, humidity 45–55%, HEPA filtered air, group housing and with access to food and water *ad libitum*. All experiments were performed in accordance with Polish law and were approved by the local Jagiellonian University Ethical Committee (licence no: 99/2018, 145/2018).

### 2.11. Induction and Monitoring of the Tumor Size in Subcutaneous Model of Human CRC

Total 35 of NOD-SCID mice at 6–8 weeks of age were used for the experiment. HCT116 cells were grown in McCoy’s 5A standard culture medium supplemented with 10% FBS and gentamicin (50 µg/mL) at 37 °C and humified atmosphere with 5% of CO_2_. One day before implantation, the medium was changed for a fresh one and avoided of antibiotics. When the confluence reached 50–60%, the cells were detached with trypsin, counted and washed in sterile PBS to remove the remnants of the culture medium. Afterwards, 1.5 × 10^6^ of HCT116 cells were injected subcutaneously into the right flank of the NOD-SCID mice, in a volume of 100 µL of PBS. Implantation of the cells was performed under laminar flow to provide sterile conditions. The growth of subcutaneous tumors was monitored every other day by external caliper measurements in two dimensions. The volume of tumor (y) was calculated according to the following formula:y = a × b^2^/2 [mm^3^]
where: a—tumor length [mm], b—tumor width [mm].

When the tumors exceeded a volume of 80 mm^3^, the mice were divided into five experimental groups and received the first dose of treatment (Day 0). Due to the spread in the tumor size, animals were randomized into the groups—each group contained animals with equal spread of the tumor size. Tumor growth was monitored by external caliper measurements, three times a week from day 0 to day 26.

### 2.12. Mice Treatment with L. lactis Bacteria

The broth cultures of *L. lactis*(hsTRAIL+) bacteria and corresponding negative control (*L. lactis*(“empty” vector)) were prepared as described above. The bacteria were induced with nisin for 2.5 h in CM, then washed with PBS supplemented with ZnSO_4_ (100 µM) to remove the remnants of the culture medium. The bacteria were injected intratumorally at a dose of 1 x 10^9^ c.f.u., in a volume of 100 µL of vehicle: PBS supplemented with ZnSO_4_ (100 µM), nisin (50 ng/mL) and aprotinin (2 µg/mL), twice a week at two consecutive days, for four weeks. At the last day of experiment, one additional dose of the bacteria was given for further evaluation of intratumor secretion of TRAIL by *L. lactis*(hsTRAIL+). The control group of mice (“Mock control”) was treated with intratumor administration of the vehicle only (100 µL).

### 2.13. Mice Treatment with MetF

MetF was administered at a dose of 125 mg/kg [44] via gastric gavage five days a week, at the same time for 25 consecutive days. Fresh MetF stock solutions were prepared in sterile PBS, filtered through a 0.22 µm filter (Carl Roth) and stored at 4 °C.

### 2.14. H&E and Immunohistochemistry Staining

When all subcutaneous tumors exceeded size of 10 mm × 10 mm (Day 26), one hour after the last dose of bacteria, mice were euthanized, and the tumors were isolated. From each experimental group, three randomly selected tumors were fixed in 10% formalin, ON. Then, paraffin-embedded tumors were sectioned (3.5 µm) and subjected to hematoxylin-eosin (H&E) staining and immunohistochemistry (IHC) analysis (from each paraffin-embedded tumor, slides for both H&E and IHC staining, were prepared). For necrotic areas analysis after H&E staining, the individual color pixels from the respective images were counted using imagemagick tool (https://imagemagick.org/, accessed on 15 April 2021) and converted. Appropriate pixels were chosen for calculating the percentage of necrotic area (N) using the pain.net software. For each treatment group, representative results are shown as mean of necrotic area (%) of the presented images. For IHC analysis, the slides were incubated with sodium citrate at 97 °C for 35 min, then incubated with appropriate antibodies. To examine the local production of TRAIL, slides were incubated with rabbit monoclonal anti-human TRAIL antibody (Cell Signalling Technology, Danvers, MA, USA) at a dilution of 1: 100 in Dako Antibody Diluent (Dako, Glostrup, Denmark). Visualisation was achieved using Dako REAL^TM^ ENVison^TM^ Detection System (Dako) according to the manufacturer’s instructions. All slides were analysed in a blinded manner using an OLYMPUS IX70 microscope and the CellSens Dimension software (Olympus).

### 2.15. Survival of L. lactis(hsTRAIL+) Bacteria within the Tumor

Survival of *L. lactis*(hsTRAIL+) bacteria in the tumors was assessed by PCR detection of the hsTRAIL-cDNA in excised subcutaneous tumors after their homogenization. Briefly, randomly selected (two mice/each group) subcutaneous tumors were cut into small pieces with a sterile scalpel, then transferred to Eppendorf tubes. Homogenization was performed in the presence of M17 culture medium and metal beads, at frequency 20 Hz for 2 min using a Tissue Lyser II (Qiagen, Hilden, Germany). The obtained homogenates were plated in a volume of 100 µL on M17 agar (1.5%) plates, in the presence of 0.5% glucose and Cm10. After 48 h of incubation at 30 °C, growing colonies were randomly isolated to prepare ON broth cultures. Plasmid DNA was isolated using the Miniprep DNA Purification Kit (Eurx) according to the manufacturer’s instructions for isolation of plasmids from gram-positive bacteria. For this reason, the isolation of plasmid DNA was preceded by digestion of the cell wall by lysozyme (10 mg/mL; Sigma Aldrich) in 10% glycerol at 37 °C for 45 min. The concentration of the isolated plasmids was measured using a Quawell Q500 spectrophotometer (Quawell Technology, Inc., San Jose, CA, USA). In order to verify the presence of the hsTRAIL-cDNA insert, PCR using primers 5’-TGGTACTCGTGGTCGTAGCA-3’ sense and 5’-GAAGCTTCGTGGTCCATGTC-3’ antisense, was performed. The PCR reaction products were subsequently separated on a 1.5% agarose gel with 0.5 µg/mL EtBr. Perfect™ 100 bp DNA Ladder (Eurx) was used as a size marker. The gels were photographed using a Gel Logic 1500 Digital Imaging System (Kodak, Rochester, NY, USA) after excitation of the fluorescence of EtBr-DNA complex with UV light. In order to create an appropriate positive control, plasmid isolated from the fresh stock (stored at −80 °C) of *L. lactis*(hsTRAIL +) was also examined.

### 2.16. Statistical Analysis

Statistical analyses were performed using GraphPad Prism software version 6.00 (GraphPad Software Inc., San Diego, CA, USA) and Microsoft Excel (version 2019, Microsoft Corporation, Redmond, WA, USA). Comparisons between multiple groups were performed using analysis of variance (ANOVA test). According to the number of dependent variables influencing the result, a one- or two-factor test was used (as indicated in the legend to the figures), with post-hoc multiple comparisons by Tukey’s method. Data were presented as mean ± SEM. The statistically significant results were showed as * *p* < 0.05, ** *p* < 0.01, *** *p* < 0.001.

### 2.17. Graphics

All graphics were created using BioRender software (BioRender.com, accessed on 15 April 2021).

## 3. Results

### 3.1. Killing of Cancer Cells in a Co-Culture of L. lactis(hsTRAIL+) Bacteria with Human CRC Cells

Previously we have documented the biological activity of *L. lactis*-derived hsTRAIL, when HCT116 cells were incubated in the presence of supernatant from the broth culture of *L. lactis*(hsTRAIL+) bacteria [41]. Here we assessed the potential anti-tumor effect of hsTRAIL-producing bacteria in a co-culture with human CRC cells. For this purpose *L. lactis*(hsTRAIL+) or corresponding control (*L. lactis*(“empty” vector)) bacteria were added to the culture of HCT116 human CRC cells, or cells of the normal colon epithelium (FHC cell line) and after 48 h of co-culture, viability of the cells was assesed by MTS assay. Data presented in Figure 1a document a significant reduction of the number of viable cancer cells co-cultured in the presence of *L. lactis*(hsTRAIL+) bacteria in a dose-dependent manner. The most pronounced anti-tumor activity of hsTRAIL-producing bacteria, demonstrated by the reduction of viable HCT116 cells to 32.9% (average) of control cultures (without bacteria), was observed for the dilution factor 1/15 of the *L. lactis*(hsTRAIL+) culture, which corresponded to MOI = 700. Similarly, the viability of another human CRC cell line, SW480, was significantly reduced in the presence of hsTRAIL-expressing bacteria (Appendix A). The co-culture of CRC cells with the bacteria containing empty vector did not affect their viability nor the co-culture of *L. lactis*(hsTRAIL+) with FHC line had any effect on viability of these normal epithelial cells (Figure 1b). The additional analysis of the expression of TRAIL-death receptors (DR4, DR5) on the surface of CRC cells and subsequent demonstration, that their blocking significantly inhibited *L. lactis*-derived-hsTRAIL induced apoptosis, confirmed the role of extracellular pathway of apoptosis in elimination of CRC cells by *L. lactis*(hsTRAIL+) (Appendix A).

### 3.2. L. lactis(hsTRAIL+) Bacteria Affect the Growth of HCT116-Spheres

The biological activity of *L. lactis*-derived hsTRAIL was subsequently verified in a 3D system using commercially available spheres of HCT116 cells. The 6-day culture of HCT116-spheres in the presence of supernatant from the broth culture of *L. lactis*(hsTRAIL+) significantly decreased their viability, when compared to the negative control (supernatant of *L. lactis*(“empty” vector), *p* < 0.001, Figure 2a), confirming an antitumor action of hsTRAIL also in a 3D culture system. However, a decrease in viability of the spheres to 74% of the control (viability of the spheres cultured in a standard culture medium), observed at a concentration of hsTRAIL of 50 ng/mL, remained at the similar level even if a 20× higher concentration of hsTRAIL was tested (81.2% viable cells in the control at the concentration of 1000 ng/mL). The presence of hsTRAIL in the culture environment of HCT116-spheres significantly affected their morphology (Figure 2b,c)—treatment of the spheres with the supernatant of *L. lactis*(hsTRAIL+) resulted in a final formation of the “small spheres” with a diameter in the range of 20.57–21.09 µm (depending on the concentration range 50–1000 ng/mL of hsTRAIL, Figure 2b), which might suggest the inhibition of clonal proliferation of a single HCT116 cell. However, when the spheres were grown in the presence of concentrated supernatant from the broth culture of *L. lactis*(“empty” vector) or culture medium alone, as controls, they formed characteristic tumor packaging with much larger diameters (97.92–50.78 µm depending on the volume of the supernatant from the control bacteria, corresponding to the TRAIL-concentration range of 50–1000 ng/mL, or 134.89 µm, *p* < 0.001, respectively). It is worth to mention that an addition of the supernatant from the culture of *L. lactis*(“empty” vector) bacteria in a volume of 380 μL, equal to the volume of *L. lactis*(hsTRAIL+) culture supernatant containing hsTRAIL in the concentration of 1000 ng/mL, resulted in significant reduction in the sphere formation, most likely due to the dilution of the growth factors in the culture medium (Figure 2b). Comparable effects to the supernatants from *L. lactis*(hsTRAIL+) cultures were observed when HCT116-spheres were grown in the presence of bacteria in a co-culture model (Figure 2d).

### 3.3. Drugs with Anti-Tumor Activity Enhance the Action of hsTRAIL Produced by L. lactis(hsTRAIL+) against Human CRC Cells In Vitro

Investigating the potential enhancement of hsTRAIL activity against HCT116 cells by drugs with proven anti-tumor activity, we demonstrated a joined effect of hsTRAIL (50 ng/mL) produced by *L. lactis*(hsTRAIL+) bacteria with 5-FU, CPT-11 and MetF (Figure 3a–c). Taking into account that combinations of Dulanermin (rhTRAIL; Genentech, Inc., South San Francisco, CA, USA) with FOLFIRI regimen (with or without bevacizumab/cetuximab/irinotecan already tested in clinical studies turned out to be not successful (NCT00671372, NCT00873756), MetF—the hypoglycaemic drug used in type II diabetes, with documented anti-tumor activity [45,46,47,48] was selected for further studies, as having an encouraging profile of joined activity, when combined with the supernatant from *L. lactis*(hsTRAIL+) broth culture (Figure 3a). In our experimental setup, MetF used at the dose of 20 mM in combination with hsTRAIL was able to reduce the viable HCT116 cells to 40.4% of control (mean value), comparing to 96.9%, when MetF was used as a single agent (*p* < 0.001). To complete the data on combined anti-tumor effect of hsTRAIL and MetF, we subsequently titrated supernatant from the broth culture of *L. lactis*(hsTRAIL+) to concentration range of 10–100 ng/mL and used in HCT116 cell culture with MetF (20 mM). The TRAIL-titration dependent effect of combined treatment on cancer cell viability was presented in Figure 3d. These results confirmed our observation regarding a joined anti-tumor effect of bacteria-produced hsTRAIL and MetF in the range of selected concentrations. Based on these observations and keeping in mind well documented safety for humans, MetF was selected for further combined CRC therapy with *L. lactis*(hsTRAIL+) bacteria in in vivo model.

### 3.4. Anti-Tumor Activity of L. lactis(hsTRAIL+) Bacteria in Subcutaneous Model of Human CRC Can Be Enhanced by MetF

Next, we evaluated the biological activity of *L. lactis*(hsTRAIL+) bacteria against cancer cells in vivo in subcutaneous mouse model of CRC. For this purpose, NOD-SCID mice (*n* = 7) with human HCT116 tumors developed subcutaneously in the right flank, were treated with *L. lactis*(hsTRAIL+) or *L. lactis*(“empty” vector) bacteria given by intratumor injections. In parallel, the potential enhancement of anti-tumor activity was evaluated by co-treatment of animals with bacteria and MetF, applied orally. A detailed scheme of the experiment, including treatment arms and a therapy-timeline is presented in Figure 4. Tumor growth was monitored three times a week by two-dimensional caliper measurements. As indicated in Figure 5a, in the mock control group receiving intratumor injections of the vehicle only, HCT116-tumors grew exponentially, increasing their size by 9.1-fold during the experiment. The tumor size was also increased in the mice receiving control bacteria (*L. lactis*(“empty” vector)), with the final 7.8-fold increase, documenting a lack of anti-tumor activity of the *L. lactis* NZ9000 strain by itself. Contrary, intratumor injections of 1 × 10^9^ c.f.u. of *L. lactis*(hsTRAIL+) or monotherapy with MetF given orally, significantly retarded the tumor growth, when compared to the mock control (*p* < 0.001, both) (Figure 5a, in black). In the *L. lactis*(hsTRAIL+) + MetF co-treated group, the tumor growth was inhibited the most, as compared to the control group (*p* < 0.001). Comparison of the tumor growth in the groups with monotherapies and combined therapy supported the enhancement of anti-tumor effect by joined action of MetF and hsTRAIL (in orange: MetF vs. MetF + *L. lactis*(hsTRAIL+), *p* < 0.05; in green: *L. lactis*(hsTRAIL+) vs. MetF + *L. lactis*(hsTRAIL+), *p* < 0.05).

Similar observations were made when the tumor growth in each group of mice was compared to the therapy with control bacteria (Figure 5a, in grey). These findings together indicate enhancement of anti-tumor action of hsTRAIL by MetF in vivo in subcutaneous model of CRC.

During the experiment, the subcutaneous tumors were also analysed macroscopically. On day 19th the appearance of extensive tumor necrotic areas was noticed in two groups of animals receiving intratumor injections of *L. lactis*(hsTRAIL+) bacteria with and w/o MetF (Figure 5b). Induction of CRC cell death in these groups by in vivo delivery of hsTRAIL was further confirmed by histological analysis of the resected tumors after H&E staining (Figure 5c). The areas of necrotic cells were also detected in tumor-sections from the animals treated with the control bacteria or MetF only, but these were much less pronounced and not detected macroscopically (Figure 5b,c).

To confirm the survival of *L. lactis*(hsTRAIL+) bacteria within the subcutaneous CRC tumors, in the next step we performed PCR analysis for hsTRAIL-cDNA on bacterial colonies isolated from the tumor tissues. Aseptically excised tumors were homogenised and plated onto M17 agar plates in the presence of Cm as a selection agent (Figure 6a). The PCR analysis of plasmids isolated from randomly picked bacterial colonies clearly demonstrated that the bacteria within the HCT116-tumors are carriers of human TRAIL-cDNA (Figure 6b). Parallel IHC staining identified intratumor location of hsTRAIL secreted by *L. lactis*(hsTRAIL+) (Figure 6c, Appendix A). Co-treatment of mice with MetF, administered orally by gastric gavage, did not affect the intratumor expression of hsTRAIL (Figure 6c, Appendix A). Simultaneously, IHC staining of tumors from other experimental groups, indicated no signal for human TRAIL (Figure 6c). The secretion of hsTRAIL within the tumor was safe for the treated animals (assessed by the regular measurements of the animal body weight, Appendix A).

## 4. Discussion

For many years TRAIL has been considered as a possible “holy grail” of anticancer therapy, due to its ability to induce apoptosis selectively in many types of cancer cells [49,50,51,52]. However, a short biological half-life in human plasma after intravenous administration led to the disappointing, low anti-tumor efficacy of rhTRAIL (Dulanermin) [53]. In addition, many cancer cells turned out to be resistant to TRAIL-induced apoptosis [54]. Thus, number of studies have been looking for the novel formulations of TRAIL (extensively reviewed by de Miguel et al. [55]) or TRAIL-based co-therapies with agents sensitizing tumor cells to its activity [30,31,32,36,37,56,57,58]. Here, to extend the time of TRAIL activity at the tumor site, we propose a local delivery and secretion of hsTRAIL by genetically modified *L. lactis* bacteria. Furthermore, to overcome the problem of cancer cell resistance to TRAIL, we propose a combined therapy model, consisting of *L. lactis*(hsTRAIL+) bacteria and metformin, as the TRAIL-potentiating agent [30].

Previously we have designed hsTRAIL-secreting *L. lactis* bacteria and have shown that such bacteria can effectively produce and secrete biologically active protein upon induction with nisin [41]. In this study we evaluated the role of *L. lactis*(hsTRAIL+) bacteria as carriers for hsTRAIL in different experimental setup of human CRC, including co-cultures of bacteria with cancer cells in vitro and subcutaneous in vivo CRC mouse model of human xenograft. In the case of in vitro studies, the cytotoxic effect of hsTRAIL-producing *L. lactis* bacteria, mediated by the binding of secreted hsTRAIL to its death- receptors DR4 and DR5 expressed on the surface of CRC cells and induction of apoptosis was demonstrated. While direct secretion of hsTRAIL in a co-culture system was cytotoxic to CRC cells, it did not affect viability of normal colon epithelium. This result is consistent with our previously published data on hsTRAIL treatment of human cardiac fibroblasts [41], confirming its safety for normal cells. A slightly different tendency in elimination of CRC cells by hsTRAIL was observed in the 3D HCT116-tumorsphere model. In cancer research, the in vitro spherical models are considered as a link between cancer cell line 2D cultures and animal in vivo models [59,60,61]. Depending on the type, they can mimic particular tumor-related parameters, such as cell heterogeneity, hypoxia or expansion [61]. In this study we showed, that the supernatant from *L. lactis*(hsTRAIL+) bacteria, or the bacteria added directly to the culture of HCT116-tumorspheres, strongly affected their morphology and led to their elimination (opposite to standard culture medium, where extensive proliferation of the cancer cells and formation of large spheres, were observed). However, the increase in hsTRAIL concentration did not correlate with further elimination of HCT116-spheres, as was detected for HCT116 cells growing in a monolayer [41]. Similar discrepancy was observed in other studies using spherical cancer models, e.g., Chandrasekaran et al. showed that the breast cancer cell lines MCF7 and BT20, after forming 3D tumor spheroids, were more resistant to TRAIL-mediated apoptosis [62]. Similarly, Vörsmann et al. showed resistance of melanoma cell spheroids to TRAIL, while the same cells grown in a monolayer were sensitive to TRAIL-induced apoptosis [63]. These observations were explained by changes in the expression level of TRAIL receptors—DR4 and DR5—as a consequence of hypoxia in the spheroid culture [62]. However, the tumor spheres belong to spherical cancer model with different characteristics than spheroids, including lack of hypoxia [61]. Our data suggest, that hsTRAIL may affect the clonal proliferation of cancer cells at their very early stage, while more investigations are needed to clarify the mechanism of this observation.

Interactions of hsTRAIL with its death receptors DR4 and DR5 is crucial for induction of apoptosis via an extracellular pathway. Therefore, to assess if anti-tumor activity of bacteria-secreted hsTRAIL is the result of its interactions with death receptors, we analysed the expression of DR4 and DR5 on the surface of untreated HCT116 cells. Flow cytometry analysis confirmed that HCT116 cells express both types of the receptors. Interestingly, an average expression of DR4 was 15.63%, while DR5 was present on the surface of 99.8% of HCT116 cells. However, their expression on the surface of CRC cells may not be an indicator of their affinity to TRAIL, nor may correlate with their role in the induction of apoptosis [52]. As further analysis of the role of binding of *L. lactis*(hsTRAIL+)-derived hsTRAIL to these receptors and their contribution in the induction of HCT116 cell apoptosis goes beyond the scope of this study, we have used a cocktail of antibodies blocking both receptors and showed their involvement in the hsTRAIL induced death of HCT116 cells (Appendix A).

The anti-tumor action of hsTRAIL-producing bacteria was further confirmed in vivo in subcutaneous model of human CRC xenografts, where *L. lactis*(hsTRAIL+) bacteria were administered intratumorally. *L. lactis* bacteria are facultative anaerobes and therefore, after intratumor administration continue growing in the tumor tissue. In this study we have proven that genetically modified *L. lactis* bacteria maintain their metabolic activity within the tumor, resulting in the local production of human recombinant protein inside the tumor mass. Secreted hsTRAIL significantly reduced the growth of HCT116-tumor within 7 days after the first administration of bacteria. So far, the intratumor delivery of TRAIL has been tested using modified *Escherichia coli* [27] or *Salmonella typhimurium* [28,29,64], both having an ability to colonize solid tumors [65,66]. In subcutaneous model of human lung cancer, *E. coli* bacteria, genetically modified to secrete human TRAIL, completely inhibited the growth of H460-tumors [27]. Delivery of TRAIL by *S. typhimurium* was examined in subcutaneous models of breast cancer [28], gastric cancer [64] and melanoma [29]. However, when *S. typhimurium* was tested in Phase I clinical trial covering the group of 24 patients with metastatic melanoma and one renal cancer patient, the expected bacterial accumulation in the tumor, following intravenous injection of the attenuated bacteria was demonstrated in three patients only [67]. Moreover, in all patients the progression of disease occurred, and one of the crucial side effects was bacteraemia [67]. Instead, *L. lactis* bacteria can serve as a promising vector for bio-delivery of therapeutic proteins in humans [68,69,70,71,72,73,74,75]. With its Generally Regarded As Safe (GRAS) status and common use as probiotics, *L. lactis* might be a valuable vehicle especially in the treatment of gut diseases. This was already proven in clinical trial of inflammatory bowel disease (IBD) treatment where *L. lactis*, as first genetically modified microorganism, was used for delivery of rhIL-10 [38]. The potential application of *L. lactis*(hsTRAIL+) bacteria for oral treatment of orthotopically developed human CRC in a mouse model is a subject of our next study (manuscript in preparation).

In the case of HCT116 subcutaneous mouse tumors treated with intratumor injections of TRAIL, cancer cell apoptosis was already documented [50]. In this study, intratumor delivery of hsTRAIL-producing bacteria led also to development of visible necrotic areas of the tumor. Approximately 1.5-times less necrotic area was observed after H&E staining of the tumor sections in the animals treated with *L. lactis*(“empty” vector) bacteria. The anticancer action of *L. lactis*, towards different cancer cell lines in vitro, including human CRC, has been suggested by Han et al. [76]. In our studies, however we did not observe any significant cytotoxic effect of *L. lactis*(“empty” vector) bacteria to human CRC cells, both in a co-culture and after treatment of HCT116 cells with the supernatant from the control bacteria broth culture (this study and [41]). Moreover, the observed level of tumor necrosis in vivo was not accompanied by the retardation of the tumor growth. Therefore, we ruled out the role of *L. lactis* bacteria growth within the tumor, as a major in elimination of cancer cells. In contrast, the observed necrosis in this group could be explained as a result of intensive cancer cells proliferation in highly hypoxic conditions, further enhanced by additional “delivery” of lactic acid [77]. Furthermore, microscopically-assessed necrotic areas after the treatment with *L. lactis* (hsTRAIL+) in monotherapy vs. in combination with MetF was not fully consistent with the results from the measurements of the tumor growth, suggesting a role of other mechanism involved, such as cancer cell proliferation rate within the tumor mass, as the impact of MetF on the proliferative capacity of cancer cells, including CRC, has been shown both in vitro and in vivo [78,79,80,81]. Another option is that the observed necrosis is secondary due to the lack of sufficient elimination of apoptotic cells within the tumor mass. This hypothesis is further supported by the presence of groups of cells with condensed nuclei, typical for apoptosis, observed in the sections of the tumors from *L. lactis*(hsTRAIL+) + MetF co-treated mice (Figure 5).

Considering the sensitivity threshold of HCT116 cells to hsTRAIL, we subsequently tested drugs, which were expected to enhance elimination of CRC cells in vivo. Although, based on in vitro data the combination of 5-FU or CPT-11 with *L. lactis*(hsTRAIL+) bacteria might lead to the reduction of the chemotherapy dose and in consequence, to the reduction of the side effects in the clinic, it has been proven that cancer cells can become resistant to chemotherapeutics, including 5-FU and CPT-11, during the treatment. Moreover, these combinations turned out to be not highly effective in clinical trials using TRAIL combined with FOLFIRI regimen (with or without Bevacizumab)/Cetuximab/Irinotecan—NCT00671372, NCT00873756). With this in mind, and based on in vitro testing, an antidiabetic drug—MetF—was chosen for the co-treatment with hsTRAIL-secreting *L. lactis* bacteria. The interest in MetF as a potential anticancer drug began in 2005, when Evans et al. published results, suggesting that type II diabetes patients treated with MetF showed a lower risk of cancer development [45]. Further, more specific epidemiological studies also confirmed a lower risk of CRC development with MetF-treated diabetes [46,47,48], while Higurashi et al. recently showed a preventive effect of MetF on metachronous colorectal adenoma/polyp formation in patients with high-risk of adenoma occurrence [82]. The mechanism of anti-tumor action of MetF is still unclear, but in vitro and in vivo studies suggest that MetF suppresses cancer development via inhibition of mitochondrial complex I, thus affecting oxidative phosphorylation in cancer cells [83,84,85]. Other results indicated that metformin increased the apoptotic rate of different cancer cells via ER stress-associated mechanisms [86,87].

Our data, however, show discrepancy in effectiveness of MetF in in vitro and in vivo settings. Although we have no formal proof, we can speculate that this observation may be due to differences both in MetF pharmacokinetics and metabolism of HCT116 cells in cell culture vs. tumor tissue. Both issues are related to the presence (or lack) of other cell types, tissue architecture, vascularization, concentration of nutrients and other factors. Because nutrients and growth factors are in “supraphysiological” concentrations in the cell culture media when compared to the tumor tissue, in fact they could reduce the HCT116 cancer cell sensitivity to MetF in vitro. Actually Birsoy et al. showed the importance of glucose concentration in sensitivity of cancer cells to biguanides [88], supporting this scenario. MetF has also been shown to restore the sensitivity of human CRC cell lines for TRAIL in vitro via degradation of antiapoptotic Mcl-1 protein, a member of the Bcl-2 family [30] and turned out to be very promising also in combination with immunotherapy with check-point inhibitors [89,90].

Here, the co-treatment of subcutaneous CRC-bearing NOD-SCID mice with *L. lactis*(hsTRAIL+) bacteria and MetF significantly retarded the tumor growth. However, the co-treatment with MetF and *L. lactis*(hsTRAIL+) resulted in far less effectiveness, without reaching a statistical significance, comparing to the results from in vitro studies. This might be due to the relatively low TRAIL production in the tumor mass detected by IHC, affecting the overall results of such a therapy. This however may be overcome in the future studies using, not-available at the time of this research, plasmid vectors, allowing to obtain a constitutive expression and secretion of the human proteins in *L. lactis* bacteria, without the need for induction (MoBiTec).

In our study, MetF was administered once a day by gastric gavage. The choice for oral administration of MetF was based on the results obtained by Dowling et al., who analysed MetF pharmacokinetics in HCT116 xenograft-bearing NOD-SCID mice [44]. While intraperitoneal (i.p.) injection of MetF resulted in only partial delivery of the drug to the tumor mass, its administration in drinking water led to continual uptake of MetF to the xenograft. In consistence, Chandel et al. showed that oral dosing of MetF resulted in a higher concentration of the drug in the tumor, when compared to its i.p. administration [91]. Regarding intratumor injection as another option for MetF administration, the study by Iliopoulos et al. on breast cancer-xenografts demonstrated, that oral delivery of MetF is equally effective in tumor regression as its injection near the tumor site [92]. Lastly, this route for MetF administration would be of relevance in case of the CRC patients treatment with *L. lactis*(hsTRAIL+) bacteria, given orally. The potential relevance of such combined therapy is a subject of our next study using mouse model of orthotopically developed human CRC.

## 5. Study Limitations

The major limitation of our therapy model presented in this paper is the use of immunocompromised NOD-SCID mice, enabling for examination of the efficacy of *L. lactis*-secreted hsTRAIL in “human system”, however at the same time, not allowing to assess the complex immune interactions between the tumor and its environment, including tumor infiltrating lymphocytes. The other limitation could be the lack of group of mice, receiving intratumor treatment with *L. lactis*(“empty” vector) bacteria and MetF via gastric gavage. Although we did not observe any significant difference in elimination of the tumor mass between the group of mice receiving monotherapy with *L. lactis*(“empty” vector) and treated with the vehicle, such a group would additionally support our results of joined effect of hsTRAIL and MetF in vivo.

## 6. Conclusions

To the best of our knowledge, this is the first study providing that *L. lactis* bacteria, harbouring a plasmid with hsTRAIL-cDNA are able for local secretion of biologically active hsTRAIL within the tumor in subcutaneous model of human CRC. This is also the first study providing evidence on the enhancing effect of MetF, given orally, on the anti-tumor activity of hsTRAIL acting locally in the tumor mass.

## Figures and Tables

**Figure 1 cancers-13-03004-f001:**
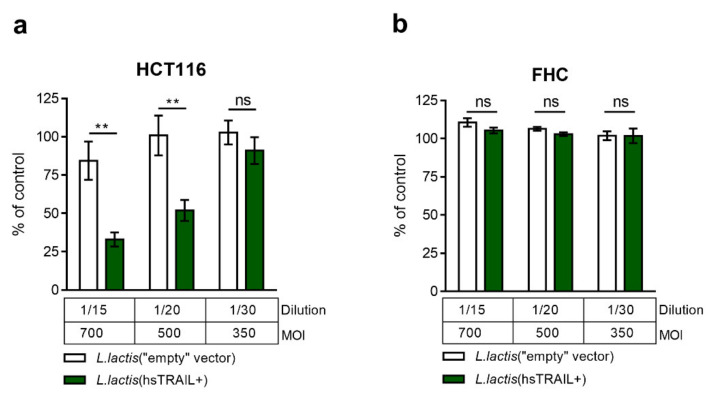
Selective cytotoxicity of *L. lactis*(hsTRAIL+)-derived hsTRAIL against colon cancer cells in a direct co-culture model of human cell lines with bacteria. Human HCT116 CRC cells (**a**) and normal epithelial colon cells (FHC) (**b**) were cultured for 48 h in the presence of *L. lactis*(hsTRAIL+) or corresponding control bacteria (*L. lactis*(“empty” vector)) with addition of nisin (inducer). Viability of cancer and non-malignant cells was assessed by MTS test. Results are showed as % of viability of cells incubated in an appropriate standard culture medium (*y*-axis) without bacteria. Legend: Multiplicity of Infection—MOI—is the number of bacteria per single eukaryotic cell; 1/15–1/30—dilution of *L. lactis*(hsTRAIL+)/*L. lactis* (“empty” vector) broth culture. The bars indicate the mean value ± SEM of three independent experiments, each performed in triplicates (*n* = 3). Statistical significance was calculated using two-way ANOVA test, with Tukey’s multiple comparisons post-hoc test. ** *p* < 0.01.

**Figure 2 cancers-13-03004-f002:**
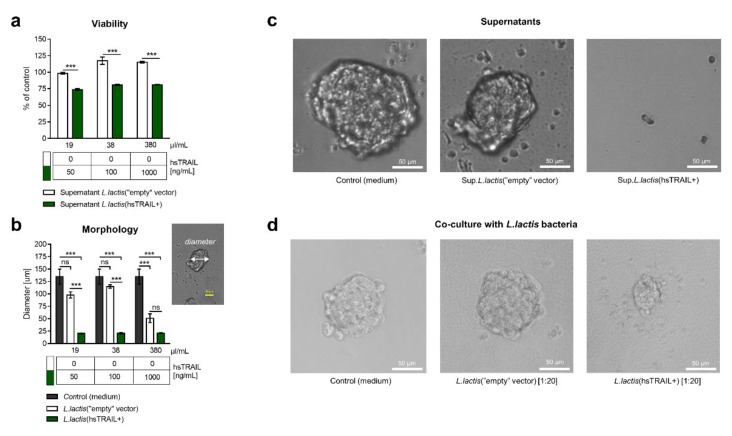
Activity of hsTRAIL against HCT116-derived tumorspheres: (**a**) HCT116-spheres were cultured for six days with increasing concentrations of hsTRAIL in supernatant from the broth culture of *L. lactis*(hsTRAIL+) bacteria. Corresponding volumes of supernatant from the broth culture of *L. lactis*(“empty” vector) bacteria was used as a negative control. The tumorsphere viability was assessed using MTS assay and shown as % of viable HCT116-spheres incubated in a standard culture medium (*y*-axis). Legend: *x*-axis—volume of the specific supernatant added/mL of a total volume of a sample; Table below—concentration of hsTRAIL [ng/mL] in a specified volume of supernatant. The bars indicate the mean value ± SEM from three independent experiments performed (*n* = 3). Statistical significance was calculated using two-way ANOVA test, with Tukey’s multiple comparisons post-hoc test, *** *p* < 0.001; (**b**) Morphology of HCT116-spheres after six days of treatment was assessed by measuring their diameter [µm]. The measurements were performed for optically the largest spheres using paint.net (version 4.1.6), as indicated in the attached photo. Legend: *x*-axis—volume of the specific supernatant added/mL of a total volume of a sample; Table below—concentration of hsTRAIL [ng/mL] in a specified volume of supernatant. The bars indicate the mean value ± SEM, from four independent experiments performed (*n* = 4). Statistical significance was calculated using two-way ANOVA test, with Tukey’s multiple comparisons post-hoc test. *** *p* < 0.001; (**c**) Representative photos of HCT116-spheres cultured for six days in the presence of a standard culture medium, supernatant from broth culture of *L. lactis*(hsTRAIL+) or *L. lactis*(“empty” vector). Brightfield microscopy, scale bar: 50 µm; (**d**) Representative photos of matured HCT116-spheres co-cultured for 48 h in the presence of standard culture medium, *L. lactis*(hsTRAIL+) bacteria in a dilution 1/20 of the broth culture or *L. lactis*(“empty” vector) broth culture at the same dilution. Brightfield microscopy, scale bar: 50 µm.

**Figure 3 cancers-13-03004-f003:**
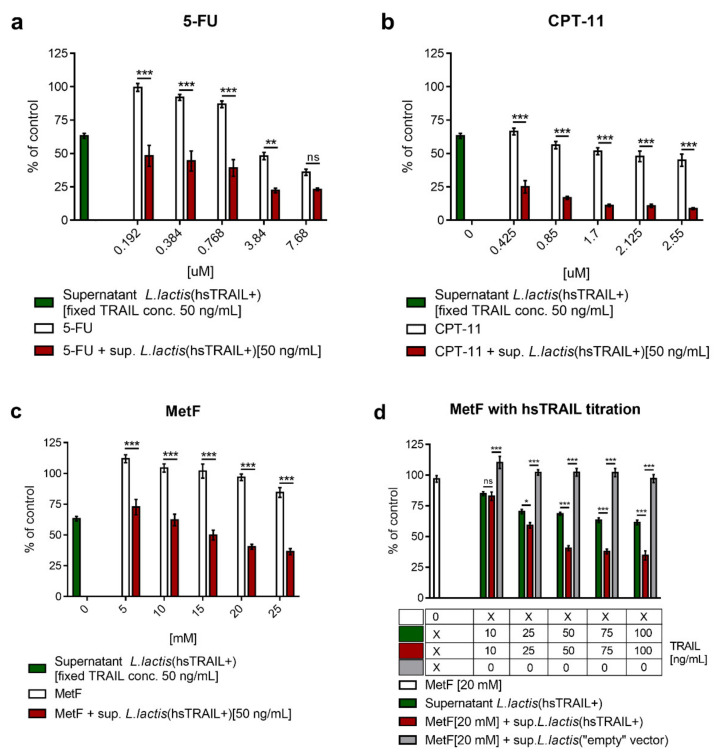
Anti-tumor activity of the selected drugs in combination with hsTRAIL against HCT116 cells in vitro. HCT116 cells were cultured for 48 h in the presence of 5-FU (**a**), CPT-11 (**b**) or MetF (**c**), in monotherapies or in combination with hsTRAIL (50 ng/mL, in a form of supernatant from the broth culture of *L. lactis*(hsTRAIL+)); (**d**) HCT116 were cultured for 48 h in the presence of fixed concentration of MetF (20 mM) and different concentrations of hsTRAIL, in a form of supernatant from the broth culture of *L. lactis*(hsTRAIL+). As negative control, supernatant from *L. lactis*(“empty” vector) bacteria was used in the corresponding volumes. Viability of HCT116 cells was assessed by MTS assay. Results are shown as % of viable cells incubated in an appropriate standard culture medium (*y*-axis). x axis—drug concentrations. The bars indicate the mean value ± SEM of three independent experiments, each performed in triplicates (**a**,**b**) or four independent experiments (**d**). Statistical significance was performed using two-way ANOVA test, with Tukey’s multiple comparisons post-hoc test. * *p* < 0.05, ** *p* < 0.01, *** *p* < 0.001.

**Figure 4 cancers-13-03004-f004:**
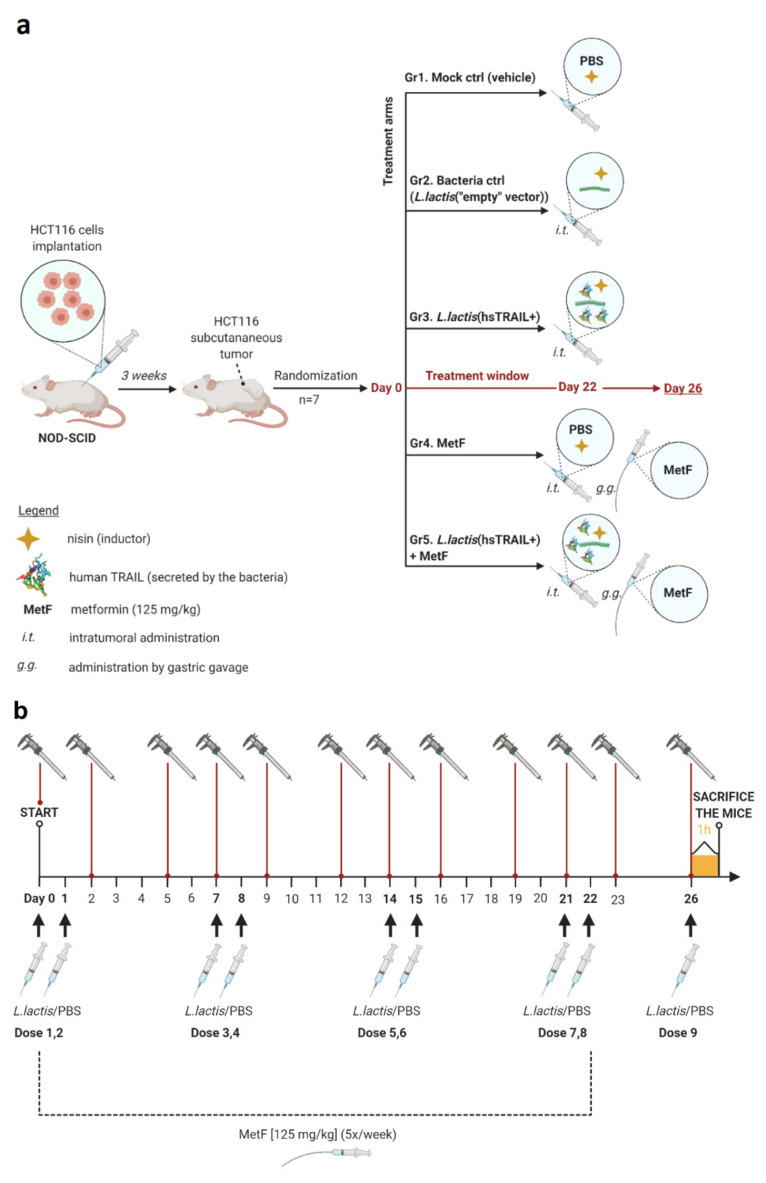
Therapy-scheme in NOD-SCID mice with subcutaneous human CRC tumors: (**a**) Experimental groups. Subcutaneous tumors of human CRC were developed by implantation of 1.5 × 10^6^ of HCT116 cells in NOD-SCID mice in the right flank. Three weeks later, animals were divided into five experimental groups (*n* = 7/group) and the treatment was started (“Day 0”), according to the following schedule: Group I (Mock control)—intratumor injections of the vehicle (PBS + ZnSO_4_ + aprotinin+ nisin; see Materials and Methods); Group II (negative control)—intratumor injections of *L. lactis*(“empty” vector); Group III—intratumor injections of *L. lactis*(hsTRAIL+); Group IV—intratumor injections of the vehicle and MetF (125 mg/kg, gastric gavage); Group V—combined therapy with *L. lactis*(hsTRAIL+) (intratumor) and MetF (125 mg/kg, gastric gavage). All intratumor solutions were supplemented with nisin at a dose of 50 ng/mL; (**b**) Timeline of the experiment. The subcutaneous tumors were monitored for 26 days from the beginning of treatment (“Day 0”) by external caliper measurements three times a week. All intratumor injections were performed twice a week for four consecutive weeks and on the last day of the experiment (arrows) to further examine hsTRAIL production within the tumor. After 1 h of the last intratumor treatment, animals were sacrificed and subcutaneous tumors were excised for further analyses. MetF was administered in Group IV and V via gastric gavage, five times/week for four following weeks.

**Figure 5 cancers-13-03004-f005:**
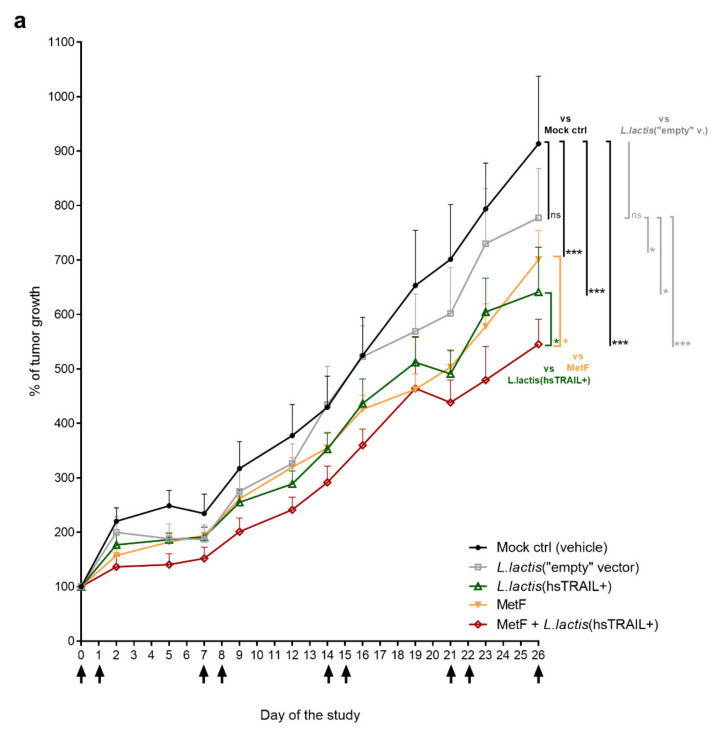
Anti-tumor activity of intratumorally administered *L. lactis*(hsTRAIL+) bacteria and MetF, given by gastric gavage: (**a**) Monitoring of the tumor size during the experiment. Tumor size was examined three times per week by external caliper measurements and calculated as described in Materials and Methods. Changes in the tumor size (y axis) are shown after normalization to the first day of the experiment (“Day 0”, designated as 100%). The arrows indicate time points of the treatment—each point indicates the mean value for *n* = 7 animals ± SEM. Statistical analysis was performed using two-way ANOVA test, with Tukey’s multiple comparisons post-hoc test. * *p* < 0.05, *** *p* < 0.001. Differences against the mock control group were marked in black, to the vector ctrl group was marked in grey; while comparison between MetF vs. co-tretament group or *L. lactis*(hsTRAIL+) vs. co-treatment group was marked in orange or green, respectively; (**b**) Macroscopical changes in subcutaneous HCT116-tumors after the treatments. The photos show representative tumors in animals from individual experimental groups, taken on day 19 of the experiment; (**c**) Histological analysis of subcutaneous tumors and quantification of the necrotic areas (H&E staining). Yellow dashed line denotes the area, which was further magnified to indicate the presence of cancer cells with condensed nuclei, characteristic for apoptosis (photo in the yellow frame). N—necrotic area; arrow—“border” of the viable tumor cells. Scale bar: 500 µm. Table—results of the necrotic area quantification (for details see Materials and Methods).

**Figure 6 cancers-13-03004-f006:**
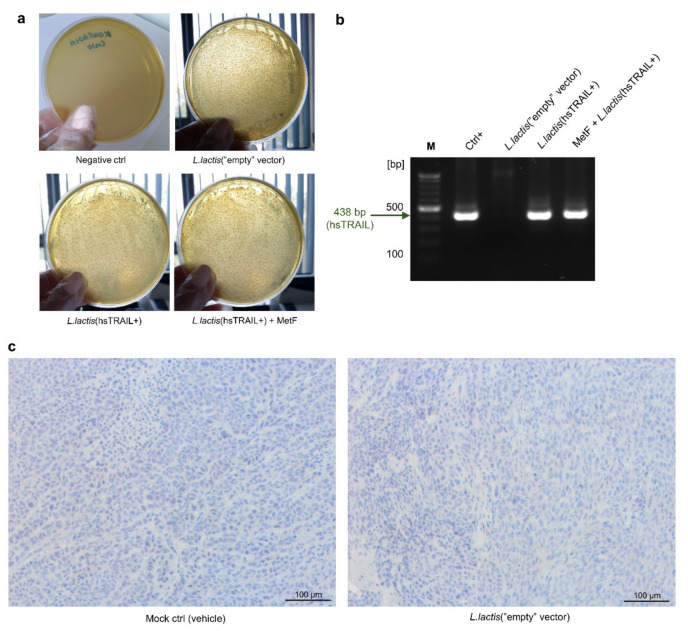
*L. lactis*(hsTRAIL+) bacteria survival within subcutaneous tumors and intratumor location of hsTRAIL produced by the injected bacteria: (**a**) Subcutaneous tumors were excised, homogenized and inoculated on plates containing M17 agar medium with Cm10 as a selection agent. Plasmids isolated from randomly picked bacterial colonies were subjected to PCR analysis, using the primers specific for hsTRAIL-cDNA sequence insert (438 bp; arrow); (**b**)Results from electrophoresis of PCR products are shown. Legend: M—size marker (bp); Ctrl+—positive control (details in Materials and Methods); (**c**) IHC staining of tumor sections for hsTRAIL protein. Representative photos of paraffin-embedded slides of tumors subjected to IHC analysis for detection of intratumor presence of human TRAIL are shown (brown spots). Scale bar: 100 µm. White dashed lines denote areas that were subsequently magnified (200%) and corrected for light intensity, sharpness/crispiness, and contrast (with the same parameters for all magnified areas; for better quality presented also in Appendix A).

## Data Availability

The data presented in this study are available in this article (and supplementary material).

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
