# Peer review of "The Anti-Tumor Effect of Lactococcus lactis Bacteria-Secreting Human Soluble TRAIL Can Be Enhanced by Metformin Both In Vitro and In Vivo in a Mouse Model of Human Colorectal Cancer"

_cancers, 2021, doi:10.3390/cancers13123004_

Round 1

Reviewer 1 Report

Several results are still problematic :

- Only two independent experiments were done with the SW480 cells (Figure 1b), which does not allow to make statistics. Moreover, at the dilution of 1/30, the supernatant of L.lactis("empty" vector) decreased the % cell viabilty of SW480 cells? How do the authors explain this result?

- Regarding Figure 2: the % viability decreases slightly after treatment with the supernatant L.lactis (hsTRAIL+) while the size of the speroids is greatly reduced. How could the authors explain these data ? Moreover, in the legend of Figure 2a, it was written that the bars indicate the mean value +/- SEM, n=3. How many experiments were done independently? The StemTek Therapeutics was founded by Angel Martin and Olatz Esnaola and was based in Bilbao, Spain and not in USA.

- In the in vivo experiment, a treatment group is still missing: L.lactis ("empty" v) + MetF. Sorry, I did a mistake in my previous report because I asked for combination of L.lactis("empty" v) with PBS which had no interest. In fact, in order to look at an enhancement of MetF treatment by intratumoral injection of L.lactis (hsTRAIL+), the control arm combining MetF with L.lactis ("empty" v) is clearly missing. It should not be forgotten that the treatment with L.lactis ("empty" v) gave 58.09% of necrosis in the tumors, which was higher than that obtained after treatment with the Mock control (PBS) (22.53% of necrosis).

- The % of necrosis measured in the tumors from mice treated with L. lactis (hsTRAIL+) alone (85,26%) was similar to the % of necrosis measured in the tumors from mice treated with L. lactis (hsTRAIL+) + MetF (83,21%), whereas the treatment with L.lactis (hsTRAIL+) alone gave 85,26 % of necrosis and the treatment with MetF alone gave 54.21% of necrosis. These data do not even allow demonstrating an additive effect of the treatment combining L.lactis (hsTRAIL+) with MetF.

Reviewer 2 Report

The manuscript after revision is definitely improved. The authors responded in a timely and precise manner to the criticisms of the reviewers, therefore I recommend its publication in Cancers.

Author Response

We thank this Reviewer for their positive opinion on our manuscript.

Reviewer 3 Report

Authors substantially corrected manuscript. I recommend article for publication.

Author Response

(The authors gave the same response as above.)

Round 2

Reviewer 1 Report

After these last revisions, the paper is acceptable for publication in its present form. 

This manuscript is a resubmission of an earlier submission. The following is a list of the peer review reports and author responses from that submission.

Round 1

Reviewer 1 Report

The authors provided new data on the use of genetically modified non-pathogenic Lactococcus lactis bacteria-secreting human soluble TRAIL as a new potential vector for local delivery of hsTRAIL in CRC. However, this paper require major revisions before being suitable for publication in Cancers.

Major points:

  •  Figures 1 a and b: Is the cell death effect induced by L.lactis(hsTRAIL+) in HCT116 and SW480 cells dependent on the binding of TRAIL on its death receptors DR4 and DR5. The authors have to show that this cell death effect can be blocked by antibodies directed against DR4 and/or DR5.
  • Figures 2 a and b: Why the cell death effect of L.lactis(hsTRAIL+) against HCT116-derived tumorspheres is not dependent on TRAIL concentrations. Is this cell death effect dependent on the binding of TRAIL to its death receptors DR4 and DR5. The authors have to show that cell death effect induced by supernatant or in conditions of co-culture with L.lactis bacteria can be blocked by antibodies directed against DR4 and/or DR5.
  • Figure 3a: to complete these data of synergistic effect of TRAIL (50 ng/ml) with increased concentrations of MetF, could the authors show the effect of the combination of MetF (20 mM) with increased concentrations of TRAIL in HCT116 cells?
  • Figure 5a: the anti-tumor activity of MetF given by gastric gavage or intratumorally administered L.lactis(hsTRAIL+) bacteria alone or in combination with MetF was compared to the mock control (vehicle) but not to the bacteria ctrl (L.lactis(“empty” vector). Could the authors compare all these treatments with the bacteria ctrl (L.lactis(“empty” vector) treatment? In fact, a control group is missing corresponding to mice bearing tumors injected with the bacteria ctrl (L.lactis(“empty” vector) and receiving PBS by gastric gavage.
  • Figure 5c: could the authors quantify necrotic areas?
  • Figure 6c: Could the authors improve the quality of the Figure 6c because the TRAIL detection on tumor sections is very difficult to see.

Author Response

.

Reviewer 2 Report

This is an interesting study testing the GMO L.Lactis in killing CRC cells via hs TRAIL system. The authors demonstrated the anti-tumor effects of GMO L.lactis in in vitro and in vivo models. In addition, combinatory therapies were also tested and MetF was selected to achieve synergistic anti-cancer effects. While the results seem to be confirming their hypothesis, some additional details would be appreciated to better help this reviewer understand the study. Please see my comments below:

For the co-culture assay, why the cells were cultured for another 30 mins after killing the bacteria, and then incubated for another 48 hours? Would this additional 48 hours wait cause additional cell death that may not related to the addition of L. lactis?

From Fig. 3, it seems that MetF by itself may even increase the cancer cell growth?

Could the authors provide some justifications for the drug dosages used in both in vitro and in vivo studies? 125 mg/Kg MetF in mice model seems to be pretty high?

How was the L. lactis treatment time of 25 days in the in vivo model determined? Why was the treatment on for two consecutive days and off for five days?

While MetF by itself seems not working well in in vitro cell model, it is interesting to see it is actually working effectively in vivo (figure 5A), do the authors know the possible explanation?

The study can be enhanced via adding additional mechanistic investigation, for example, can the authors confirm the cell death are induced exactly the way they think (e.g., apoptosis vs. necrosis)?

The conclusion /or end of the discussion section should also talk about the study limitations.

Author Response

.

Reviewer 3 Report

The manuscript of Kaczmarek et al presents an interesting study on the in vitro and in vivo antitumor activity of Lactococcus lactis bacteria-secreting 2 human soluble TRAIL in colon carcinoma. The studio is well designed and well executed and deserve publication in Cancers after minor revision.

  I have only a few observations:

1) Although the efficacy of metformin in colon cancer is well illustrated by the authors in the discussion, its use with respect to 5-FU and CPT-11 is unclear. The latter two drugs, unlike metformin, are used in clinical practice. As shown in Figure 3, these two drugs at all concentrations used induce an increase in cytotoxicity in the presence of the supernatant of L. Lactis (hsTRAIL). Therefore, the combined use could consequently reduce the doses of chemotherapy and therefore also the side effects.

In addition, the concentrations of metformin in the range 10-25 mM, to the best of my knowledge I do not think they are concentrations that can be obtained in plasma.

2) If the effect is synergistic in combined therapy, it should be proven by analysis with appropriate models of synergy (e.g. Chou-Talalay).

3) In figure 3 the levels of cytotoxicity induced by the sup. L. Lactis (hsTRAIL) should be shown.

4)Also in figure 3 the concentrations of 5-FU and CTP-11 are expressed in microM and not in microg / ml.

Author Response

.

Reviewer 4 Report

Many bacteria have impact on development of colorectal cancer. The most important are anaerobic Fusobacterium sp., and Porphyromonas sp. Please see and cite https://www.mdpi.com/2076-2607/7/1/20 in the Introduction. Simultaneously, many bacteria have anti-cancer activity. Authors of the reviewed article presented anti-tumor effect of Lactococcus lactis. Studies were made both in vitro on human colon carcinoma cell lines and in vivo in a mouse model of human colorectal cancer. Studies are very good presented, both methodology and results. Authors showed a significant reduction of the CRC tumor growth by injection of L. lactis producing hsTRAIL. Because Lactobacillus belongs to GRAS bacteria (Generally Recognized as Safe), presented studies can have practical aspect in cancer treatment. Moreover, using of L. lactis seems safe in most patients. I recommend this article for publication after adding of information about carcinogenic and anti-cancer bacteria in Introduction. Additionally, figures 5a, 5c and 6c should be larger and better quality.

Author Response

.

Reviewer 5 Report

Kaszmarek et al. report on a bacterial system that can produce human TRAIL for localized production of this tumoricidal protein, with in vitro data to show cytotoxic activity and add-on of combination with various small molecule drugs, and a final in vivo xenografted HCT116 tumor model. The data could potentially become promising, but there are essentially no controls included, the rationale for combinatorial treatment is not defined, and the response in the in vivo model is less than overwhelming. In general, the data seems haphazardly generated without a setup to answer a predefined goal/hypothesis.

Major comments:

To determine whether the cytotoxicity is truly TRAIL-mediated one would at a minimum expect a control with non-induced L. Lactis (hs TRAIL+)  bacteria as well as with recombinant human TRAIL and TRAIL neutralizing antibody and or caspase-8 inhibitor to validate the TRAIL-dependency of the effect. In addition, the measurement of toxicity is solely based on MTS and may be convoluted by live bacteria that can also convert MTS. Therefore, TRAIL-mediated activity should at a minimum be also established with other types of assays, such as activation of caspases (e.g. using fluorometric probes or western blotting), apoptosis assays such Annexin-V/PI or DNA fragmentation. Again, specific neutralization of TRAIL activity in these settings would give strength to the conclusion on the TRAIL-dependency of the effect. These type of controls are lacking throughout the manuscript.

Major comment 2:

The proof of efficacy data provided with 2 CRC cell lines is minimal, especially as data with HCT116 was also previously described in ref 22, with the therapeutic effect very quickly tapering off to n.s. at an MOI of 350. The therapeutic window of treatment seems very limited. POC studies should be performed with a panel of cell lines and ideally with primary CRC cells. In this setting, it would be worthwhile to perform the spheroid experiments as organoids from primary CRC material in which the cancer stem cell impact be delineated.

Major comment 3:

For most if not all in vitro data, the experiments were only performed twice, so the statistical analysis presented in the paper is inappropriate. All experiments need to be done in 3 independent replicates (not 2 as indicated).

Major comment 4:

It would seem that for a more realistic assessment of the effect of L.Lactis (rhTRAIL) a more representative trans-well or even trans-epithelial model (with monolayer of e.g. MDCK or Caco-2) with bacteria is warranted. In patients, the bacteria will be in the colon and TRAIL will need to pass the epithelial barrier. It is questionable how feasible/efficient this would happen.

Major comment 3:

In the synergism graphs the treatment with only bacteria is lacking, making any conclusion on ‘synergy’ impossible.

Major comment 4:

In Figure 2A, there is only a marginal (and dose-independent) effect of hsTRAIL. These data do not fit with the earlier experiments. Indeed, MTS not be fully representative of the effect as there is a much clearer impact on diameter. This could be growth inhibitory effect or direct cytotoxic effect, which should be determined. Why was the analysis shifter from MTS to diameter from supernatant in A/C vs. B/D bacteria.

In addition, the better experiment is to treat already formed spheroids and analyse the impact of L. Lactis in this setting (preferably in a trans-well system). Finally, these studies should be performed in multiple cell lines and ideally organoids of primary CRC.

Major comment 5:

The HCT116 xenograft data seems to be missing from the paper.

Major comment 6.

The therapeutic effect in the primary CRC model is rather limited, with no statistical analysis provided between EV L. lactis and TRAIL-containing L. Lactis (by eye this would likely be not statistically significant. Similarly, the combination effects with metformin is only compared to untreated conditions, which is not fair. Combination should be compared to both single treatment and if the reduction in tumor size of the sum of both treatments is indeed less than that of the reduction by combination treatment, one could make an argument for synergy. However, looking at the data this would not be the case.

Major comment 7

In general, the mouse experimental data seems very limited in scope. One would expect to have caspase-3 stainings in the tumor material, to have toxicity data on all organs and a graph mouse weight, as well as to have data on the plasma levels of rhTRAIL.

Major comment 8

It was claimed that the bacteria continue growing regardless of the hypoxia in the tumor tissue. However, this data is based on bacteria extracted from tumor tissue after 1 hour of injection, which to me seems an unlikely early time-point for bacteria to potentially all die. Longer timeslot would be more accurate to confirm it. 
